# *Giardia lamblia* risk factors and burden in children with acute gastroenteritis in a Nicaraguan birth cohort

Lester Gutiérrez[1], Nadja A. Vielot [2]*, Roberto Herrera[1], Yaoska Reyes[3], Christian Toval-Ruíz[4], Patricia Blandón[5¤], Rebecca J. Rubinstein[3], Javier Mora[1,6], Luther A. Bartelt[7,8,9], Filemón Bucardo[2,6], Sylvia Becker-Dreps[2,3], Samuel Vilchez[2,5]

1 Centro de Investigación de Enfermedades Tropicales (CIET). Facultad de Microbiología. Universidad de Costa Rica, San José, Costa Rica, 2 Department of Family Medicine, University of North Carolina at Chapel Hill, Chapel Hill, North Carolina, United States of America, 3 Department of Epidemiology, University of North Carolina at Chapel Hill, Chapel Hill, North Carolina, United States of America, 4 Universidad Tecnológica La Salle, León, Nicaragua, 5 Department of Microbiology and Parasitology, National Autonomous University of Nicaragua-León, León, Nicaragua, 6 Laboratory of Helminthology, Faculty of Microbiology, University of Costa Rica, San José, Costa Rica, 7 Division of Infectious Diseases, Department of Medicine, University of North Carolina at Chapel Hill, Chapel Hill, North Carolina, United States of America, 8 Department of Microbiology and Immunology, University of North Carolina at Chapel Hill, Chapel Hill, North Carolina, United States of America, 9 Center for Gastrointestinal Biology and Disease and the Departments of Medicine, University of North Carolina at Chapel Hill, Chapel Hill, North Carolina, United States of America

¤ Current address: Faculty of Medicine, Medical Technology, Universidad Nacional Comandante Padre Gaspar García Laviana, León, Nicaragua
* nadjavielot@unc.edu

## Abstract

### Background

*Giardia lamblia* is an intestinal protozoan estimated to cause ~200 million symptomatic infections annually, mainly in children in low- and middle-income countries associated with intestinal damage, increased permeability, and malabsorption.

### Methods and results

We describe here the epidemiology, incidence, clinical characteristics, and risk factors of acute gastroenteritis episodes (AGE) with *G. lamblia* detection (GAGE) using a birth cohort of 443 Nicaraguan children followed weekly until 36 months of life. From June 2017 to July 2021, 1385 AGE samples were tested by qPCR. *G. lamblia* was detected in 104 (7.5%) of AGE episodes. In all, 69 (15.6%) children experienced at least one GAGE episode, and 25 of them (36.2%) experienced more than one episode. The incidence rate of the first episode of GAGE was 6.8/100 child-years (95% CI, 4.5–9.1). During GAGE, bloody stools, vomiting, and fever were uncommon, and children were less likely to be treated at a primary care clinic, suggesting that GAGE is typically mild and most cases did not receive medical attention, which could facilitate higher parasite loads with increased possibilities of establishing chronic carriage. GAGE was more common in children 12–24 months of age (13.9/100 child-years [95% CI, 10.7–17.1]) as compared to other age groups. In our birth-cohort, children living in a home with an indoor toilet (aHR, 0.52 [95%CI, 0.29–0.92]), and being

**Data Availability Statement:** Deidentified data are available in S1 Dataset.

**Funding:** This work was supported by the National Institute of Allergy and Infectious Diseases

(R01AI127845 to LG, NAV, RH, YR, CT-R, PB, FB, SBD, SV; R01AI151214 to LAB; K24AI141744 to SBD) and the Fogarty International Center (LG, RH, YR) of the National Institutes of Health. The funders had no role in study desing, data collection and analysis, decision to publish, or preparation of the manuscript.

**Competing interests:** The authors have declared that no competing interests exist.

breastfed in the first year of life (aHR: 0.10 [95%IC, 0.02, 0.57]) had a lower incidence of GAGE. In contrast, being breastfed for $\leq 6$ months was associated with a higher incidence if the children were living in houses without indoor toilets and earthen floors (HR, 7.79 [95% CI, 2.07, 29.3]).

## Conclusion

Taken together, GAGE is more frequent under poor household conditions. However, breast-feeding significantly reduces the incidence of GAGE in those children.

## Author summary

*G. lamblia* is an intestinal protozoan having mixed associations with acute gastroenteritis, yet positive associations with impaired child growth. *G. lamblia* infects millions of people worldwide annually and is more prevalent in children residing in low- and middle-income countries, making it an important target for prevention and control efforts. Although *G. lamblia* carriage is widespread, its epidemiology, natural history, genetic diversity, host genetic susceptibility factors, and post-infectious sequelae in children are still under investigation. As no human vaccine is available, understanding its risk factors in an endemic country is the first step for reducing the burden and mitigating its sequelae. In a birth cohort in León, Nicaragua, we found that children living in households with poor sanitation conditions are at higher risk of AGE with *G. lamblia* detection (GAGE), but prolonged duration of breastfeeding despite these conditions may reduce the risk in these children. The study highlights the importance of breastfeeding and the need to better understand how *G. lamblia* contributes to the global burden of disease in children, and the mechanisms of host protection to guide future interventions.

## Introduction

*Giardia lamblia* (also known as *Giardia duodenalis* or *Giardia intestinalis*) is an enteric protozoan that is considered one of the most common agents of intestinal infection worldwide with around 200 million estimated infections annually. It is most prevalent in children in low- and middle-income countries (LMIC) [1,2]. *G. lamblia* is transmitted through the fecal-oral route and infections can vary from asymptomatic to acute diarrhea, which may progress to chronic episodes [3,4], but *G. lamblia* detection does not appear to be associated with moderate-to-severe diarrhea in LMIC [5]. Instead, in developed countries, this parasite has been reported in clustered daycare outbreaks [6] and is characterized by seasonal and recreational waterborne transmission [7,8]. Additionally, it is one of the most common causes of diarrhea in travelers [9]. Although *G. lamblia* carriage is widespread, its natural history, genetic diversity, host genetic susceptibility factors, clinical characteristics, and post-infectious sequelae in children like intestinal damage, increased intestinal permeability, and inadequate nutrient absorption [10–13], are still under investigation.

In children from LMIC, the incidence of GAGE varies across different sites [13]. The Etiology, Risk Factors, and Interactions of Enteric Infections and Malnutrition and the Consequences for Child Health and Development Project (MAL-ED) multisite birth cohort study conducted in 8 countries, found a global aggregate incidence of 15.0 GAGE per 100 child-years (95% CI: 13.8–16.3), with a higher incidence in the second year of life [13]. Reducing the

incidence of *G. lamblia* in highly endemic areas remains a challenge, mainly due to the recent increases in antiparasitic drug resistance [14,15] and the absence of an available vaccine. However, vaccine candidates are being investigated [16]. In addition, Water, Sanitation, and Hygiene (WASH) improvements and nutrition interventions have been widely recommended [17–20], but are of uncertain effect in reducing *G. lamblia* acquisition or disease.

There is little information about the burden and risk factors for *G. lamblia* in Central America. For León, Nicaragua, the most recent study that includes *G. lamblia* dates back more than a decade, prior to widespread use of molecular methods for *G. lamblia* detection [21]. Thus, we describe the risk factors for GAGE in order to guide efforts to reduce the burden of disease in children by analyzing samples and data from the Sapovirus-Associated Gastro-Enteritis (SAGE) birth cohort study from Nicaragua [22]. A secondary aim was to determine the epidemiological and clinical characteristics of GAGE updates.

## Methods

### Ethics statement

The study was approved by the Ethical Committee for Biomedical Research of the National Autonomous University of Nicaragua (UNAN) at León (Acta number 2–2017) and the Institutional Review Board of the University of North Carolina at Chapel Hill (study number 16–2079). Written informed consent to participate in the cohort and provide biological samples for storage and future unspecified analyses was obtained from the caregivers on behalf of their children.

### Participants

A total of 443 Nicaraguan children were enrolled in a population-based birth cohort study investigating childhood gastroenteritis (SAGE) in León, Nicaragua between June 12, 2017 and July 31, 2018 [22]. In brief, children were visited weekly in their households from birth until 36 months of age to surveil for AGE episodes. Furthermore, epidemiological characteristics were collected at baseline and updated at each study visit, mothers were asked whether they had breastfed their child the previous day and whether the child had consumed anything besides breast milk in the past week. Additionally, each month, mothers were asked to provide extensive risk factor data regarding household conditions and interpersonal contact, including factors that were unlikely to change on a weekly basis (e.g., water treatment measures, water storage, and the presence of animals in the household), those variables collecting are self-report and they could be subject to reporting bias. Clinical characteristics were also documented for each reported episode of AGE. Written informed consent was requested from a parent or legal guardian from each participant, and they provided their consent prior to enrollment in the study.

### Samples

Stool samples were collected from each of the reported AGE episodes as described previously [22] within 10 days of the onset of symptoms. Aliquots of 1:10 (mg/ml) stool suspension in phosphate-buffered saline (PBS) were prepared and stored at -20˚C until processing. In this cohort, 1,498 diarrhea episodes (defined as increase in stool frequency of at least three stools per 24-hour period or a substantial change in stool consistency [bloody, very loose, watery], following at least three symptom-free days) were reported, for which 1,385 stool samples were collected. Stool samples were not available if they could not be collected within 10 days of symptom onset. Furthermore, to evaluate potential host genetic factors, blood and saliva

samples were collected to determine ABO blood groups, Lewis phenotypes, and secretor status. These samples were stored at -20˚C until processing.

## Molecular detection of *G. lamblia*

To identify *G. lamblia* using Real-Time PCR, 200µl of the 1/10 stool suspension was used to extract DNA using the QIAamp Fast DNA Stool Mini Kit (Cat No./ID: 51604) and by following the manufacturer's instructions. Stool suspension was initially treated with acid-washed glass beads (0.5 mm; Sigma) and vortexed for 2–5 min, as described by Stroup SE *et al*[23], to increase cyst lysis and DNA extraction.

Real Time-PCR was performed to identify *G. lamblia* in diarrhea stool samples using the protocol described by Verweij JJ *et al* that target the small subunit ribosomal (SSU) gene (18S-like) from *G. lamblia* (GenBank accession no. M54878)[24]. In brief, 0.2M of the forward and reverse primers and probe (*G. lamblia* -80F 5'-GAC GGC TCA GGA CAA CGG TT-3', *G. lamblia* -127R 5'-TTG CCA GCG GTG TCC G-3', *G. lamblia* -105T FAM-5'-CCC GCG GCG GTC CCT GCT AG-3') were added to a PCR reaction mix consisting of 3 µL of DNA, 12.5 µL of Bio-Rad iQ Multiplex Powermix (Bio-Rad Laboratories, Hercules, CA, USA) and nuclease-free water to a final volume of 25µl. PCR conditions were 95˚C 10 min and 45 cycles: 95˚C 10 s, 60˚C 1 min (signals reading). Real-time PCR was performed using the Bio-Rad CFX96 Touch Real-Time PCR Detection System. The real-time PCR was considered positive if the cycle threshold (Ct) was of (Ct) ≤35. Carryover contamination was controlled by using nuclease-free water, during DNA purification and real-time PCR. The positive control was ATCC Quantitative Synthetic DNA from *Giardia lamblia* PRA-3006SD with a Ct of 30.

## Molecular detection of co-pathogens in GAGE

Seventy-six GAGE episodes were tested by qPCR for 13 other common enteric pathogens using oligonucleotide primers described by Liu J. *et al*[25] in the multiplex qPCR platform. These included rotavirus, adenovirus, astrovirus, norovirus, sapovirus, Enterotoxigenic *Escherichia coli* (ETEC), Enteropathogenic *E. coli* (EPEC), *Enteroinvasive E. coli/Shiguella* (EIEC/*Shiguella*), *Enteroaggregative E. coli* (EAEC), Shiga Toxin-producing *E. coli* (STEC), *Campylobacter* spp, *Entamoeba histolytica*, and *Cryptosporidium parvum*.

## ABO group, Secretor, and Lewis phenotyping

The ABO group was determined in blood samples by hemagglutination testing. In a subset of 28 children, a blood sample was not provided; therefore, blood ABO phenotyping was carried out by using the saliva sample as by described Nordgren J. *et al*[26]. Secretor and Lewis phenotyping were performed in saliva collected and processed by an in-house enzyme-linked immunosorbent assay as described previously by Reyes, *et al* [27].

## Statistical analysis

Baseline characteristics of children from the cohort were described as percentages or medians with interquartile ranges (IQR). We compared the clinical characteristics of GAGE versus AGE without *G. lamblia* detection. Then, we calculated the incidence rate (episodes/100 child-years) of the first GAGE in the cohort. Next, we determined the relative hazard and 95% confidence interval (CI) of children experiencing a GAGE episode by baseline risk factors using a Cox proportional hazards model. We also assessed the relative hazard and 95% CI of GAGE by time-varying risk factors and exposures using a time-varying Cox model. A multivariable analysis was performed using significant variables in crude analysis and was adjusted including

potential confounders for each variable. Finally, to determine whether breastfeeding duration has a protective impact on reducing GAGE in children living in a high-risk population characterized by household conditions variables that were statistically significant in the crude analysis (no indoor toilet, earthen floor, and presence of mice), we use breastfeeding duration according to the number of weeks the mother reported any breastfeeding, exclusive or not, in the first 6 months. Then, we classified children as receiving breastfeeding during at least 90% of weeks in the first six months of life "≥6 months of breastfeeding" or less than 90% of weeks in the first 6 months of life "<6 months of breastfeeding", based on the World Health Organization recommendation for six months of exclusive breastfeeding [28,29]. 45 children with less than six months of follow-up were excluded from this analysis. Statistical analyses were conducted in SAS version 9.4 (SAS Institute, Cary, North Carolina) and images were performed using GraphPad Prism V7 (GraphPad Software Inc).

## Results

Children included in the cohort were 51.0% male, 45.4% born by cesarean section, and received a median of 11.0 (IQR: 3.9–22.6) months of non-exclusive breastfeeding (Table 1 and S1 Fig). Exclusive breastfeeding was short-lived in this cohort (median of 3.1 weeks, IQR: 0.5, 5.7). Most households (84.0%) had piped municipal water, 72.5% had an indoor toilet, and 30.2% had earthen floors at home. 44.5% percent of children did not have basic needs met, according to a socioeconomic indicator created by Peña, *et al.* for this setting[30] (Table 1). By the end of the cohort, 109 children had dropped out (retention rate of 75.4%), with the highest dropout rate occurring during the first year (68/443 children) (S2 Fig).

### Incidence and Clinical Characteristics

The 443 children experienced 1,497 episodes of AGE, of which 1,385 (92.5%) were tested for *G. lamblia* by qPCR. One hundred four (7.5%) of the 1385 AGE episodes were positive for *G. lamblia*. Of these, 60 GAGE (57.7%) occurred between 12–24 months of age (median: 20.5 months; IQR: 17–26 months]). The 104 GAGE episodes were detected in 69 children (15.6%), of which 25 of them (36.2%) experienced more than one GAGE, and 5 (7.3%) experienced more than three episodes in the first three years of life. The overall incidence of having a first GAGE was 6.8 cases per 100 child-years (95% CI, 4.5–9.1) (Fig 1). Incidence was higher in children of 12–24 months of age (13.9/100 child-years [95% CI, 10.7–17.1]) compared to children younger than 12 months (2.0/100 child-years [95% CI,0.7–3.3]) or older than 24 months (6.3/100 child-years [95% CI,4.0–8.5]).

The presence of at least one other pathogen was detected in 61/76 GAGE episodes (80.3%) (Fig 2). Bacteria pathogens were detected in 70.5% and viral pathogens in 62.3%, with 32.8% having both viral and bacterial co-infections (Fig 2A). The most common co-infections were enterotoxigenic *Escherichia coli* (ETEC) (27.6%), norovirus (23.7%), and *Campylobacter* spp (18.4%) (Fig 2B). Other pathogenic parasites were not detected. In 15 (19.7%) GAGE episodes, *G. lamblia* was the only pathogen detected. Mono-infections with *G. lamblia* were more likely to be found without other co-pathogens in the first GAGE detection (Fig 2A).

The clinical profile of GAGE was characterized by a median of 5 days with diarrhea (IQR: 3–9), and a maximum of 5 stools per day (IQR: 4–6). Bloody stools, vomiting, and fever were not commonly found in GAGE (Table 2). There were no differences in clinical characteristics of AGE episodes with and without *G. lamblia* detection; however, GAGE episodes were 51.0% less likely to receive care at primary care clinic (OR, 0.49; 95% CI, 0.28–0.77) compared with AGE without *G. lamblia* detection (Table 2). Similarly, no differences in clinical characteristics were found when comparing GAGE co-infected with another pathogen versus GAGE without

**Table 1. Baseline characteristics of children in the birth cohort in León, Nicaragua.** (n = 443).

| Characteristic | n (%) or median (IQR) |
|---|---|
| **Birth and Genetic Characteristics** | |
| Gender (%male) | 226 (51.0) |
| Mode of delivery (%cesarean) | 201 (45.4) |
| Mean age of mother years at child's birth | 24.5 (20.7–28.4) |
| Blood ABO group (n = 440): | |
| Type A | 80 (18.2) |
| Type B | 37 (8.4) |
| Type AB | 5 (1.1) |
| Type O | 318 (72.3) |
| Child´s Rh factor positive (n = 415) | 402 (96.9) |
| Child´s Lewis Antigens (n = 440): | |
| Lewis a+b- | 36 (8.2) |
| Lewis a-b+ | 319 (72.5) |
| Lewis a-b- | 85 (19.3) |
| Child´s Secretor status (n = 440) | 385 (87.5) |
| **Nutrition factors (breastfeeding)** | |
| Exclusive breastfeeding (in weeks)[§] | 3.1 (0.5–5.7) |
| Non-exclusive breastfeeding (in months) | 11.0 (3.9–22.6) |
| Breastfeeding duration: | |
| <6 months | 132 (29.8) |
| 6 to 12 months | 93 (21.0) |
| >12 months | 218 (49.2) |
| **Household Conditions & Socioeconomic Indicators** | |
| Piped municipal water at home | 372 (84.0) |
| Presence of earthen floor at home | 134 (30.2) |
| Presence of indoor toilet | 321 (72.5) |
| Mother completed any secondary or high education (n = 373) | 97 (26.0) |
| Mother employed at time of child´s birth (n = 373) | 64 (17.2) |
| Basic household needs[⊥] (n = 389) | |
| Unsatisfactory | 173 (44.5) |

[§] Exclusive breastfeeding defined as the child's age in which child did not consume outside formula, foods, or water beyond breastmilk.

[⊥] Poverty index was determined according Peña *et al* [30]

co-infection. However, the duration of diarrhea in GAGE tended to be longer in the absence of other etiologies (median of days of diarrhea [IQR]: 7 [3–11] vs 4 [3–7]) (S1 Table).

## Risk factors of AGE with *G. lamblia* detection

Static and time-varying risk factors were investigated among children who had at least one GAGE (n = 69) versus children with no reported AGE with *G. lamblia* detection during cohort surveillance (n = 374). This analysis was limited to the first GAGE episodes detected. In bivariate analysis, we found that earthen floors (HR, 1.99 [95% CI, 1.23, 3.20]) and the presence of mice in the home (HR, 3.17 [95% CI, 1.43, 7.00]) were associated with a higher risk of experiencing GAGE (Table 3). Conversely, having an indoor toilet in home (HR, 0.41 [95% CI, 0.24, 0.70]), use of hand sanitizer (HR, 0.22 [95% CI, 0.09, 0.52]) and the presence of other animals in the home (including the presence of dog, cat) (HR, 0.42 [95% CI, 0.26, 0.68]) were

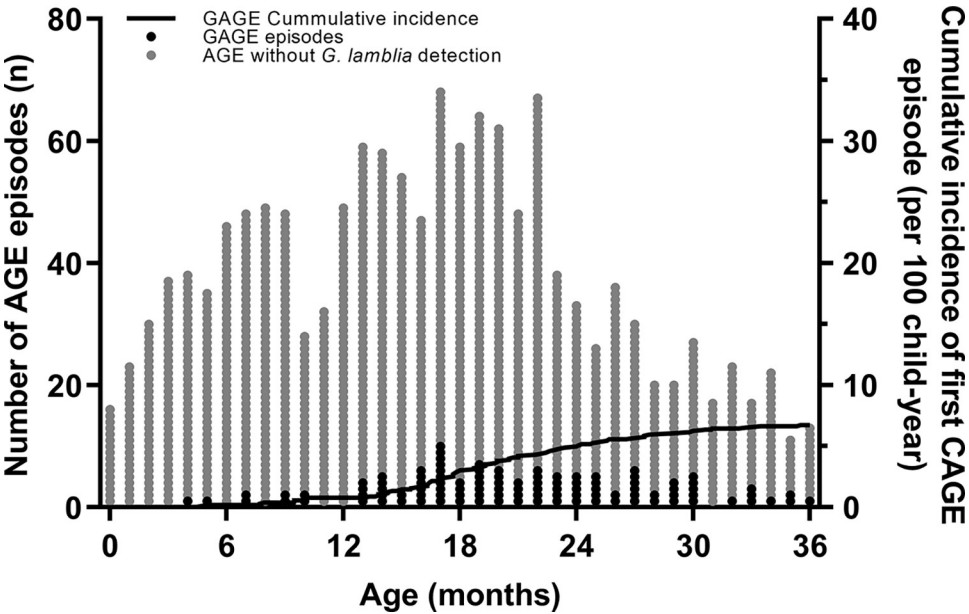

**Fig 1. Incidence and frequency of *G. lamblia* detection in acute gastroenteritis episodes (GAGE) in a birth cohort until 3 years of age.** Number of AGE episodes reported by month (left axis), no-GAGE (gray points), and GAGE (black points) in 1385 diarrhea stools. Incidence rate for first GAGE detection in 443 children followed up 36 months (right axis).

protective (Table 3). Furthermore, we found that breastfeeding in the first year of life had a protective effect against GAGE (HR, 0.17 [95% CI, 0.03, 0.84]); nevertheless, this protective effect was not observed in the second and third years of life (Table 3). Interpersonal contact (including attending daycare in the last week, a social event, playing with a child outside the

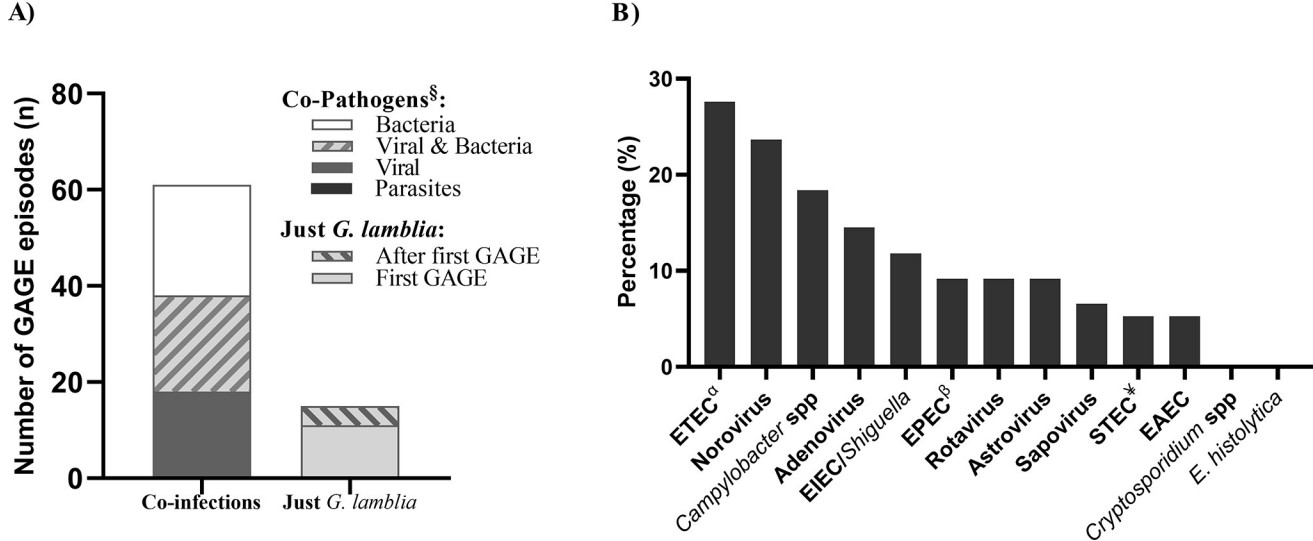

**Fig 2. Co-pathogens detected in acute gastroenteritis episodes with *G. lamblia* detection (GAGE) (n = 76).** A) Co-pathogens stratified by bacteria, virus and parasite detections (n = 61) and Mono-infection with *G. lamblia* group was stratified by the first GAGE. B) Percentage of co-infection by etiology. §All pathogens were detected by qPCR. ETEC: Enterotoxigenic *Escherichia coli*, EPEC: Enteropathogenic *E. coli*. EIEC: *Enteroinvasive E. coli*, EAEC: *Enteroaggregative E. coli*. STEC: Shiga Toxin-producing *E. coli*. αInclude both heat-labile and heat-stable toxins. βInclude both typical and atypical strains. ¥Include both Shiga-toxin (SXT) STX-1 and STX-2.

**Table 2. Clinical characteristics of Acute Gastroenteritis episodes with *G. lamblia* detections compared with AGE without presence of *G. lamblia* in a Birth Cohort of Children in León, Nicaragua.**

| Clinical Characteristic | AGE with *G. lamblia* (n = 104)[⊥] | AGE without *G. lamblia* (n = 1281)[⊥] | OR (IC95%)[†] |
|---|---|---|---|
| Days of diarrhea | 5 (3–9) | 5 (3–8) | - |
| Maximum number of stools per 24 hours | 5 (4–6) | 5 (4–6) | - |
| Blood in stool | 1 (1.0) | 61 (4.8) | 0.19 (0.27–1.41) |
| Vomiting | 20 (19.2) | 337 (26.3) | 0.68 (0.40–1.10) |
| Fever | 33 (31.7) | 504 (39.3) | 0.72 (0.47–1.10) |
| Received care at primary care clinic | 20 (19.2) | 432 (33.7) | **0.49 (0.28–0.77)** |
| Received care at hospital | 5 (4.8) | 135 (10.5) | 0.42 (0.17–1.07) |
| Hospitalization | 1 (1.0) | 56 (4.4) | 0.21 (0.03–1.56) |
| Received zinc | 27 (26.0) | 396 (30.9) | 0.79 (0.50–1.23) |
| Received intravenous fluid | 2 (1.9) | 49 (3.8) | 0.49 (0.12–2.06) |
| Received medications for gastroenteritis[‡] | 48 (46.2) | 643 (50.2) | 0.80 (0.57–1.27) |

[⊥]Data shown in n(%) or median (IQR)

[†]Binary outcomes analyzed using logistic models to estimate ORs

[‡]Include Antibiotics, symptom-relieving agents and anti-helminthic. AGE: Acute Gastro-Enteritis

home, or using public transportation in the past week), ABO blood group, secretor, and Lewis phenotypes were not statistically associated with GAGE. In addition, GAGE tended to be more frequent in males (HR, 1.28 [95% CI, 0.80, 2.05]) (Table 3).

After adjustment for potential confounders identified through analysis of causal diagrams, the associations for earthen floor, the presence of mice in the home, hand sanitizer use, and eating uncooked fruit/vegetables were attenuated or lost precision in the 95% confidence intervals. However, breastfeeding in the first year of life (aHR, 0.10 [95% CI, 0.02–0.57]), indoor toilet (aHR, 0.52 [95% CI, 0.29–0.92]), and presence of cats and dogs (aHR, 0.54 [95% CI, 0.33–0.89]) in the home remained associated with a high protective against GAGE episodes (Table 3).

### Association of breastfeeding on AGE with *G. lamblia* detection in a high-risk population

The breastfeeding duration was evaluated for reducing GAGE in children living in a high-risk population characterized by household conditions variables that were statistically significant in the crude analysis (no indoor toilet, earthen floor, and presence of mice). We found that less than 6 months of breastfeeding increased the risk of GAGE in children living in a house without an indoor toilet (HR, 3.95 [95% CI, 1.32, 11.84]) (Fig 3A), but not statistical effects was observed for the earthen floor compared to children with more than 6 months of breastfeeding (HR, 2.06 [95% CI, 0.79, 5.38]) (Fig 3B). We also found an increased risk of GAGE in children breastfed for less than 6 months living in houses that both have no indoor toilet and having earthen floors with children compared with children with less than 6 months of breastfeeding living under the same conditions (HR, 7.79 [95% CI, 2.07, 29.3]) (Fig 3C). Furthermore, there were no differences in the presence of mice in the home by breastfeeding category (Fig 3D).

### Discussion

We examined the risk factors, burden, and clinical characteristics for GAGE in order to guide efforts to reduce the burden of disease in children from a Nicaraguan birth cohort, followed

**Table 3. Static and time-varying risk factors for *Giardia lamblia* detections in a birth cohort of children in León, Nicaragua (n = 443).**

| Characteristic | Crude | | Adjusted* | |
|---|---|---|---|---|
| | HR | 95% CI | HR | 95% CI |
| **Biological factors** | | | | |
| Mode of delivery (% cesarean) | 1.08 | 0.68, 1.74 | | |
| Gender (%Male) | 1.28 | 0.80, 2.05 | | |
| Blood ABO group (n = 440) | | | | |
| Type B | 1.11 | 0.40, 3.06 | | |
| Type A | 0.85 | 0.46, 1.57 | | |
| Type O | 0.83 | 0.49, 1.40 | | |
| Type AB | 0.36 | 0.04, 3.00 | | |
| Child´s Lewis Antigens negative (n = 440) | 1.13 | 0.62, 2.08 | | |
| Child´s no Secretor status (n = 440) | 1.01 | 0.50, 20.4 | | |
| Child´s Rh factor positive (n = 415) | 0.92 | 0.21, 3.96 | | |
| Child phenotype | | | | |
| Le-Se- | 1.40 | 0.44, 4.43 | | |
| Le-Se+ | 1.04 | 0.53, 2.05 | | |
| Le+Se+ | 0.98 | 0.59, 1.72 | | |
| Le+Se- | 0.82 | 0.36, 1.88 | | |
| **Nutrition** | | | | |
| Child breastfed[¥] | **1.79** | **1.10, 2.90** | 0.97 | 0.58, 1.60 |
| First year of life | **0.17** | **0.03, 0.84** | **0.10** | **0.02, 0.57** |
| Second year of life | 1.74 | 0.94, 3.21 | 1.71 | 0.93, 3.16 |
| Third year of life | 1.04 | 0.34, 3.21 | 1.05 | 0.34, 3.24 |
| Ate uncooked fruit/vegetable[¥] | **0.33** | **0.15, 0.73** | 0.72 | 0.31, 1.69 |
| Ate outside the home[¥] | 1.40 | 0.87, 2.26 | | |
| Shared a bottle or cup with another person[¥] | 1.19 | 0.69, 2.04 | | |
| **Household Conditions** | | | | |
| Water sources (% no piped water in the home) | 0.91 | 0.48, 1.74 | | |
| Floor type (% earthen) | **1.99** | **1.23, 3.20** | 1.47 | 0.83, 2.59 |
| Sanitation type (% indoor toilet) | **0.41** | **0.24, 0.70** | **0.52** | **0.29, 0.92** |

(*Continued*)

**Table 3.** (Continued)

| Characteristic | Crude | | Adjusted* | |
|---|---|---|---|---|
| | **HR** | **95% CI** | **HR** | **95% CI** |
| Cats or dogs present in the home[¥] | **0.42** | **0.26, 0.68** | **0.54** | **0.33, 0.89** |
| Pigs present in the home[¥] | 0.37 | 0.09, 1.49 | | |
| Mice present in the home[¥] | **3.17** | **1.43, 7.00** | 1.51 | 0.68, 3.36 |
| Used hand sanitizer[¥] | **0.22** | **0.09, 0.52** | 0.67 | 0.26, 1.76 |
| **Interpersonal Contact** | | | | |
| Attended daycare, a social event, played with a child outside, or used public transportation[¥] | 0.98 | 0.47, 2.06 | | |
| Went swimming[¥] | 1.39 | 0.51, 3.82 | | |
| Presence of other child using diapers[¥] | 1.55 | 0.77, 3.11 | | |
| Had contact with anyone with AGE[¥] | 1.02 | 0.89, 1.16 | | |

*Adjusted analyses were not performed for non-significant variables. Adjusted models included potential confounders for each variable identified through analysis of causal diagrams. Breastfeeding was adjusted for child's age, supplemental feeding, and eating food outside the house; eating uncooked fruits or vegetables was adjusted for breastfeeding, child's age, and eating food outside the house; floor type was adjusted for water source and sanitation type; sanitation type was adjusted for floor type and water source; presence of animals and mice were adjusted for floor type, water source, and child's age; hand sanitizer was adjusted for water source, sanitation type, and child's age.
[¥]Time-varying risk factors (corresponds to data reported in the week prior to assessment of the outcome).
AGE: Acute Gastroenteritis. HR Hazard Ratio. CI: Confidence Interval.

until 36 months of age. We found that 7.5% of AGE episodes were associated with *G.lamblia* infection and an incidence rate of the first GAGE episodes of 6.8 per 100 child-years. In 2014, very similar rates of 8.0% and 8.8 per 100 child-years was reported in the same study area [21], indicating stable transmission patterns. In our study, GAGE episodes were detected in 15.6% of children, and 36.2% of these had more than one episode. Children with GAGE were less likely to seek care at a primary care clinic as compared to AGE episodes due to other etiologies (even when public health care is available). This finding is consistent with the association between *G. lamblia* detection and less severe diarrhea in the GEMS study [5]. We consider that the lack of care seeking for GAGE episodes could be important in the epidemiology of GAGE. If children do not seek medical attention, they may not receive a diagnosis of GAGE, potentially resulting in a lack of treatment. This absence of intervention could allow the parasite to persist, which might lead to chronic episodes. The persistence of *G. lamblia* is commonly associated with long-term sequelae[13,31,32]. Additionally, we found that the lack of access to indoor toilets and breastfeeding for less than 6 months are risk factors for GAGE.

Although we did not find differences in clinical characteristics between GAGE with and without co-infections, the duration of diarrhea tended to be longer in the absence of other etiologies. *G. lamblia* is more associated with prolonged/persistent diarrhea[3] and the impact of GAGE, in the presence of other etiologies, on the clinical characteristics and persistence of diarrhea is a critical area of research, especially in populations with high exposure to multiple enteric pathogens. Co-infections with GAGE and other pathogens do not generally increase the severity of diarrhea. The interaction *G. lamblia* with other pathogens can influence clinical

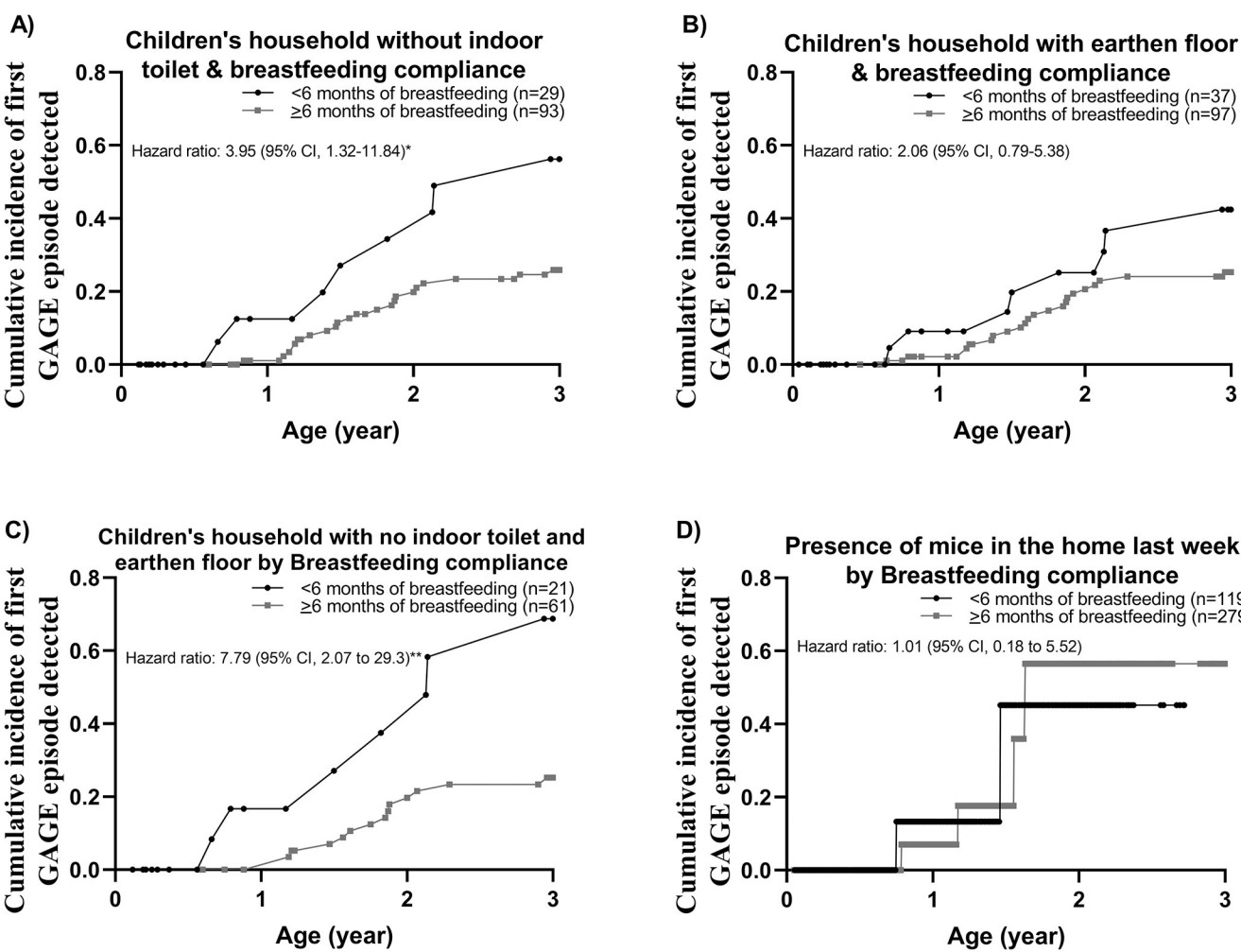

**Fig 3. Cumulative incidence and hazard ratios of first GAGE detection and along the first 3 years of life in children with presence of risk factors associated stratified by breastfeeding compliance (determined according to the number of weeks the mother reported any breastfeeding, exclusive or not, in the first 6 months).** Children were classified as receiving breastfeeding during at least 90% of weeks in the first six months of life "≥6 months breastfeeding" or less than 90% of weeks in the first 6 months of life "<6 months breastfeeding" living in: A) household without an indoor toilet (n = 122). B) household with earthen floor (n = 134). C) household that both have no indoor toilet and having earthen floor (n = 82). D) presence of mice in the home (n = 398). AGE: Acute Gastro-Enteritis. GAGE: Acute Gastro-Enteritis with presence of *G. lamblia*. ¥Time-varying risk factors (corresponds to data reported in the week prior to assessment of the outcome). *P<0.050 to 0.010, **P<0.010 to 0.001, ***P<0.001.

outcomes; for example, co-infection with rotavirus and *G. lamblia* can reduce the severity of rotavirus-induced diarrhea[33]. Moreover, *G. lamblia* can modulate host immune responses, potentially affecting the clinical outcomes of co-infections, which may influence the severity of AGE[34]. Further research is needed to fully understand these complex interactions and their implications for clinical outcomes during GAGE.

It is well known that *G. lamblia* cysts (infecting stage) are environmentally stable, being able to survive for a prolonged time, especially in moist settings [2,35]. Those characteristics facilitate the transfer of cysts from one host to another. In that sense, it is not surprising that household condition factors were associated with GAGE in our study, where unsatisfactory household characteristic like no indoor toilets at home was positively associated with GAGE. This finding is very similar to those reported previously, where parasitic infections are associated with poverty and inadequate water quality and sanitation systems[20,36,37]. Although

zoonotic transmission of *G. lamblia* has been reported [2,38–42], our finding that animals like dogs and cats were protective factors supports a limited risk of zoonotic transmission of *G. lamblia* from these animals in our population. Interestingly, in our cohort, GAGE was more common in households with presence of mice. While some *G. lamblia* strains can persist in experimental murine models [43], there is not evidence that mice are a reservoir for humans. Mice naturally carry *G. muris* and/or different *G. lamblia* genotypes (assemblage G)[40]; unfortunately, the genotyping of GAGE samples was not performed in this study. While observational studies are needed to investigate if mice can be infected with *G. lamblia* assemblages A and B to contribute to local endemicity, here we consider that the presence of mice at home is most likely be an indicator of poor sanitary conditions in the household.

Strategies to reduce the burden of *G. lamblia* infections in highly-endemic populations remain a challenge. Considering there is not human vaccine available, interventions such as health education, improved environmental sanitation, and personal hygiene programs have been widely recommended to decrease *G. lamblia* exposure[19,20,44,45]. Durable changes to the environmental structure in LMICs can be costly and take years to implement. Breastfeeding protection, however, may be the most important measurement to protect against GAGE in children under 3 years.

Breastfeeding has been well documented to protect infants from diarrhea caused by intestinal pathogens, including protozoan infections[46–48]. In our population where exclusive breastfeeding was uncommon, we found that non-exclusive breastfeeding during the first year of life significantly reduced the risk of GAGE, Moreover, non-exclusive breastfeeding during the first 6 months significantly reduced the risk of GAGE in children living in high-risk household environments (households that lack an indoor toilet and have earthen floors) compared to children breastfed less than 6 months. This finding demonstrates the importance of sustained breastfeeding in long-term protection against GAGE, and supports the significant impact of breastfeeding on reducing early infections [13]. Unfortunately, we could not determine why mothers stopped breastfeeding earlier in this cohort, and data supports the need for future qualitative and biological research to understand if breastmilk components such as IgA antibodies are important or whether the impact of breastmilk on microbiome composition may play a protective role in *G. lamblia* pathogenesis throughout the first years of life.

The Lewis and secretor phenotypes are susceptibility factors for enteric viral and bacterial infections [27,49,50]. In the current study, we found that children's ABO blood type, Lewis, and secretor phenotypes were not associated with GAGE, thus suggesting that the presence of those antigens, in the intestinal epithelia or secretions might not be associated with GAGE, as observed for some viral and bacteria gastrointestinal infections. Although genetic susceptibility was not associated with GAGE in this study, more studies are needed to understand if those antigens play a role during *G. lamblia* infections, not only in the development of GAGE but also in establishing colonization in asymptomatic events. The genetic susceptibility at individual/host levels is a topic of ongoing research.

This study has some limitations. First, this work was limited to GAGE cases and those results represent symptomatic individuals. A comprehensive study of the natural history of *G. lamblia* in this cohort would need to include asymptomatic *G. lamblia* infections which are reportedly the majority of cases [25,32]. Second, breastfeeding consistency was assessed via weekly maternal reports, which may be subject to recall or social desirability biases. Finally, some time-varying risk factors were missing if a weekly visit was missed or if the survey respondent did not know or chose not to report a response to some of the risk factor questions.

In summary, our study provides updated evidence for risk factors and burden for GAGE episodes in a Central American setting. This is a first step to reduce the disease burden in this

endemic area. Our analysis suggests that GAGE is more frequent under poor household conditions, and breastfeeding was significantly associated with the protection of children living in a high-risk environment for exposure. These findings might guide future studies to understand *G. lamblia* infections in children and interventions to prevent its impacts on childhood health and complications associated with this parasite.

## Supporting information

**S1 Table. Clinical characteristics of Acute Gastroenteritis Episodes with *G. lamblia* detections (GAGE) with and without the presence of other etiologies.**
(XLSX)

**S1 Fig. Frequency of non-exclusive breastfeeding and number of active children by age in months (n = 443 children).**
(TIF)

**S2 Fig. Frequency of dropouts throughout the SAGE cohort study (n = 443 children).** A) Frequency and retention rate over the cohort by year. B) Number of monthly acute gastroenteritis episodes (AGE) reported (gray bar, left Y axis) and total number of active children by age in months (black line, right Y axis).
(TIF)

**S1 Dataset. Deidentified Dataset containing the epidemiological and clinical characteristics of GAGE in the Nicaraguan birth cohort.**
(XLSX)

## Acknowledgments

Authors appreciate the support from all parents of the children participating in the SAGE cohort and recognize all efforts from the fieldwork team (Yorling Picado, Nancy Corea, Mileydis Soto, Maria Mendoza, Merling Balmaceda, Jhoseling Delgado, Ruth Neira, Veronica Pravia, Yuvielka Martinez, Aura Scott, Yadira Hernandez, Xiomara Obando y Patricia Mendez) for collection of clinical and epidemiological information together with biological material. The support from the SILAIS-León in particular the personnel from the Perla Maria Norori Health Center. Additionally, we appreciate the support of the University of Costa Rica (UCR) and the Doctoral Program in Sciences at UCR under the project "Cooperation with the University of North Carolina for training and scientific dissemination in infectious diseases in Central America", which has provided guidance and training for LG during his doctoral program.

## Author Contributions

**Conceptualization:** Lester Gutiérrez, Luther A. Bartelt, Filemón Bucardo, Sylvia Becker-Dreps, Samuel Vilchez.

**Data curation:** Roberto Herrera, Yaoska Reyes, Christian Toval-Ruíz, Patricia Blandón, Filemón Bucardo, Samuel Vilchez.

**Formal analysis:** Lester Gutiérrez, Nadja A. Vielot.

**Funding acquisition:** Luther A. Bartelt, Filemón Bucardo, Sylvia Becker-Dreps, Samuel Vilchez.

**Investigation:** Lester Gutiérrez, Nadja A. Vielot, Roberto Herrera, Yaoska Reyes, Christian Toval-Ruíz, Patricia Blandón, Rebecca J. Rubinstein, Luther A. Bartelt, Filemón Bucardo, Sylvia Becker-Dreps, Samuel Vilchez.

**Methodology:** Yaoska Reyes, Rebecca J. Rubinstein, Luther A. Bartelt, Filemón Bucardo, Sylvia Becker-Dreps, Samuel Vilchez.

**Project administration:** Christian Toval-Ruíz, Filemón Bucardo, Sylvia Becker-Dreps.

**Resources:** Javier Mora, Luther A. Bartelt, Filemón Bucardo, Sylvia Becker-Dreps, Samuel Vilchez.

**Software:** Nadja A. Vielot.

**Supervision:** Nadja A. Vielot, Yaoska Reyes, Javier Mora, Luther A. Bartelt, Filemón Bucardo, Sylvia Becker-Dreps, Samuel Vilchez.

**Validation:** Christian Toval-Ruíz, Samuel Vilchez.

**Visualization:** Lester Gutiérrez.

**Writing – original draft:** Lester Gutiérrez, Rebecca J. Rubinstein, Luther A. Bartelt, Filemón Bucardo, Sylvia Becker-Dreps, Samuel Vilchez.

**Writing – review & editing:** Lester Gutiérrez, Nadja A. Vielot, Roberto Herrera, Yaoska Reyes, Rebecca J. Rubinstein, Luther A. Bartelt, Filemón Bucardo, Sylvia Becker-Dreps, Samuel Vilchez.

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
