## [Decision Letter · Decision Letter 0]

16 Aug 2024

Dear Dr. Vielot,

Thank you very much for submitting your manuscript "Giardia lamblia burden and risk factors in children with acute gastroenteritis in a Nicaraguan birth cohort" for consideration at PLOS Neglected Tropical Diseases. As with all papers reviewed by the journal, your manuscript was reviewed by members of the editorial board and by several independent reviewers. The reviewers appreciated the attention to an important topic. Based on the reviews, we are likely to accept this manuscript for publication, providing that you modify the manuscript according to the review recommendations. 

Sincerely,

Paul J. Brindley, PhD

Editor-in-Chief

Paul Brindley

Editor-in-Chief

Well written manuscript. Congratulations

Reviewer's Responses to Questions

**Key Review Criteria Required for Acceptance?**

**Methods**

-Are the objectives of the study clearly articulated with a clear testable hypothesis stated?

-Is the study design appropriate to address the stated objectives?

-Is the population clearly described and appropriate for the hypothesis being tested?

-Is the sample size sufficient to ensure adequate power to address the hypothesis being tested?

-Were correct statistical analysis used to support conclusions?

-Are there concerns about ethical or regulatory requirements being met?

Reviewer #1: -Missing asymptomatic giardiasis is a huge limitation to the current works making interpretation very difficult.

-Descriptive statistics of the missing 113 diarrheal episodes (or the reason in which weekly sampling couldn’t collect a sample within 10 days) would be helpful in determining if there is bias here. 

-a CT <37 seems high. Can authors give an explanation of how this cutoff for positivity was chosen?

-How many children were excluded for having <6months of follow up

-It is unclear which variables were included in the multivariate models? All variables listed in table 3 or just those significant in univariate analysis?

-Methods detailing the collection of all of the data that makes up table 3 needs to be included in the methods. For instance, how was presence of mice in the home determined?

-Data on the quantity or frequency of breastfeeding would be helpful as so few mothers were exclusively breastfeeding and breast milk seems protective.

Reviewer #2: Study design is appropriate but, excludion of other major intestinal infections has not been done.

Sample size is sufficient and correct analysis has been done

Reviewer #3: -Are the objectives of the study clearly articulated with a clear testable hypothesis stated?

Yes, it is good to make a study of this type in order to see what the effetcs are of Giardia infections in children in this region of the world.

-Is the study design appropriate to address the stated objectives?

Yes, it is. As the authors state it would be good to look at asymptomatically Giardia infected children as well and do genotyping.

-Is the population clearly described and appropriate for the hypothesis being tested?

Yes

-Is the sample size sufficient to ensure adequate power to address the hypothesis being tested?

Yes

-Were correct statistical analysis used to support conclusions?

Yes

-Are there concerns about ethical or regulatory requirements being met?

No

**Results**

-Does the analysis presented match the analysis plan?

-Are the results clearly and completely presented?

-Are the figures (Tables, Images) of sufficient quality for clarity?

Reviewer #1: -3 weeks of exclusive breast feeding is extremely short in a LMIC setting. It would be helpful for the reader to understand if this was being supplemented with formula and, if so, the water source of households becomes very important to this study. 

-In figure 1 it would be helpful to have the sample size by month on the figure or some discussion of attrition so that the reader understands the decrease in GAGE during the 3rd year of life is not due to less children enrolled.

-Discussion of testing for other pathogens should be moved to the methods section. Why were only 76/104 GAGE episodes tested? Which other pathogens were assayed for and the methods need to be detailed.

-Lines 202-205: This is not clear? Was duration longer without co-infection? It a poorly written sentence grammatically which obscures meaning. 

-Please list p values in table 3

-Lines 233 – 239 need to moved to the methods section. They also need to be revised for clarity. The model that was created (and why the covariates were chosen as they were) should be clarified. It seems several models were created?

-Results of the breast feeding analysis are confusing. Presence of an indoor toilet is used in two of the models (one alone and one in the subset that also had an earthen floor)? Why? How do you interpret this when earthen floor is not significant?

Reviewer #2: 1.Abdominal pain and cramp is an important clinical characteristic features of AGE and G. Lamblia. According to one study , G. lamblia significantly contributes to recurrent abdominal pain in children (Buch N.A., Ahmad S.M., Ahmed S.Z., Ali S.W., Charoo B.A., Hassan M.U. Recurrent abdominal pain in children. Indian Pediatr. 2002; 39:830–834.). This was not mention in the because their main aim is to describe the clinical characteristics.

2.How long do the symptoms persist in case of AGE with positive with G. lamblia? As the symptoms are mild and moderate, do not require medication mostly and which could increase the parasitic burden lead to chronic carriage. 

3.The study did not compare the differences between malnourished vs normal children who are suffering from AGE associated with G. Lamblia? A study conducted in Pakistan indicate that between 9% and 10% of population has G. lamblia, with malnourished children under 10 years of age particularly affected. (Al-Mekhlafi M.H., Azlin M., Nor Aini U., Shaik A., Sa’iah A., Fatmah M.S., Ismail M.G., Ahmad Firdaus M.S., Aisah M.Y., Rozlida A.R., et al. Giardiasis as a predictor of childhood malnutrition in Orang Asli children in Malaysia. Trans. R. Soc. Trop. Med. Hyg. 2005; 99:686–691. doi: 10.1016/j.trstmh.2005.02.006)

4.Study did not show any relationship poverty, family size and income? But these are also a risk factor for G. lamblia.

Reviewer #3: -Does the analysis presented match the analysis plan?

Yes.

-Are the results clearly and completely presented?

Yes, well-presented.

-Are the figures (Tables, Images) of sufficient quality for clarity?

Yes

**Conclusions**

-Are the conclusions supported by the data presented?

-Are the limitations of analysis clearly described?

-Do the authors discuss how these data can be helpful to advance our understanding of the topic under study?

-Is public health relevance addressed?

Reviewer #1: -Again, is there evidence for the claim that not seeking care leads to prolonged shedding and sequelae?

-Line 266: Earthen floor was not associated with GAGE as reported above. Please clarify.

-Discussion of why breastfeeding in the 2nd year of life is not protective when almost 50% of women continued to do it would be helpful.

-I don’t think the authors can state that any queried variable is “not associated with Giardia infection” as they only tested symptomatic infections and the majority of cases are asymptomatic. Lines 299 – 303 are an example.

Reviewer #2: Yes, conclusion is supported by the data presented. It would have been nice if other socioeconomic conditions have been taken into account for analysis.

Reviewer #3: -Are the conclusions supported by the data presented?

Yes

-Are the limitations of analysis clearly described?

Yes, also discusssed in the Discussion part.

-Do the authors discuss how these data can be helpful to advance our understanding of the topic under study?

Yes.

-Is public health relevance addressed?

Yes.

**Editorial and Data Presentation Modifications?**

Reviewer #1: The grammar and sentence structure (mostly run-on sentances) is poor in places which obscures meaning.

Reviewer #2: (No Response)

Reviewer #3: Line 28. “200 million symptomatic infections”

Line 32, Explain AGE

Line 72. “G. lamblia detection does not appear to be associated with moderate-to-severe diarrhea”, this is true in low income countries but in high income countries it is associated with diarrhea.

Line 167. 45.4% born by cesarean section. High or normal int his environment?

**Summary and General Comments**

Reviewer #1: Authors have performed a secondary analysis to determine risk factors for gastroenteritis with Giardia detected and to determine the effects of breastfeeding on such episodes. The cohort was designed to study Sapovirus and thus did not include asymptomatic stool collection as the majority of Sapovirus infections are symptomatic. This is a major limitation of the current work as the majority of Giardia infections are asymptomatic. This makes interpretation and extrapolation of the presented findings extremely difficult. Identified “cases” are likely a vast under-representation of the burden of Giardia infection. That said, the findings are relatively consistent with the literature on risk factors for Giardia and the protective effect of breast feeding. This study does not add greatly to the literature as similar findings are thoroughly reported in the literature.

Reviewer #2: As above

Reviewer #3: This study essentially verifies what has already been shown in earlier studies of Giardia infections in children. However, this is a new region and setting and the conclusions are clear.

Genotyping of the GAGE samples would have been good to resolve issues of animal/assemblage infections and it could have been correlated to other data.

PLOS authors have the option to publish the peer review history of their article (what does this mean?). If published, this will include your full peer review and any attached files.

Reviewer #1: No

Reviewer #2: Yes: Rashidul Haque

Reviewer #3: No

Figure Files:

Data Requirements:

Reproducibility:

References

---

## [Decision Letter · Decision Letter 1]

22 Oct 2024

Dear Dr. Vielot,

We are pleased to inform you that your manuscript 'Giardia lamblia risk factors and burden in children with acute gastroenteritis in a Nicaraguan birth cohort' has been provisionally accepted for publication in PLOS Neglected Tropical Diseases.

Best regards,

Sarman Singh, MD, FRSC, FRCP

Academic Editor

Paul Brindley

Editor-in-Chief

Reviewer's Responses to Questions

**Key Review Criteria Required for Acceptance?**

**Methods**

-Are the objectives of the study clearly articulated with a clear testable hypothesis stated?

-Is the study design appropriate to address the stated objectives?

-Is the population clearly described and appropriate for the hypothesis being tested?

-Is the sample size sufficient to ensure adequate power to address the hypothesis being tested?

-Were correct statistical analysis used to support conclusions?

-Are there concerns about ethical or regulatory requirements being met?

Reviewer #3: The questions raised by reviewers have been address and the limitations of the study has been clarified.

**Results**

-Does the analysis presented match the analysis plan?

-Are the results clearly and completely presented?

-Are the figures (Tables, Images) of sufficient quality for clarity?

Reviewer #3: The questions raised by reviewers have been address and the limitations of the study has been clarified.

**Conclusions**

-Are the conclusions supported by the data presented?

-Are the limitations of analysis clearly described?

-Do the authors discuss how these data can be helpful to advance our understanding of the topic under study?

-Is public health relevance addressed?

Reviewer #3: The questions raised by reviewers have been address and the limitations of the study has been clarified.

**Editorial and Data Presentation Modifications?**

Reviewer #3: Accept.

**Summary and General Comments**

Reviewer #3: (No Response)

PLOS authors have the option to publish the peer review history of their article (what does this mean?). If published, this will include your full peer review and any attached files.

Reviewer #3: No

---

## [Editor Report · Acceptance letter]

5 Nov 2024

Dear Dr. Vielot,

We are delighted to inform you that your manuscript, "*Giardia lamblia* risk factors and burden in children with acute gastroenteritis in a Nicaraguan birth cohort," has been formally accepted for publication in PLOS Neglected Tropical Diseases.

Best regards,

Shaden Kamhawi

co-Editor-in-Chief

Paul Brindley

co-Editor-in-Chief
